# Ridge Augmentation Using β-Tricalcium Phosphate and Biphasic Calcium Phosphate Sphere with Collagen Membrane in Chronic Pathologic Extraction Sockets with Dehiscence Defect: A Pilot Study in Beagle Dogs

**DOI:** 10.3390/ma13061452

**Published:** 2020-03-23

**Authors:** Jungwon Lee, Yong-Moo Lee, Young-Jun Lim, Bongju Kim

**Affiliations:** 1Department of Periodontology, One-Stop Specialty Center, Seoul National University Dental Hospital, Seoul 03080, Korea; jungwonlee.snudh@gmail.com; 2Department of Periodontology and Dental Research Institute, School of Dentistry, Seoul National University, Seoul 03080, Korea; ymlee@snu.ac.kr; 3Department of Prosthodontics and Dental Research Institute, School of Dentistry, Seoul National University, Seoul 03080, Korea; 4Dental Life Science Research Institute & Clinical Translational Research Center for Dental Science, Seoul National University Dental Hospital, Seoul 03080, Korea

**Keywords:** bone regeneration, ridge augmentation, alloplast, synthetic bone, biphasic calcium phosphate

## Abstract

This study was conducted to comparatively examine spontaneous healing versus ridge augmentation, in surgically-created dehiscence defects, associated with chronic pathology in dogs. Mandibular second, third and fourth premolars (P2, P3 and P4) were hemi-sectioned, and a dehiscence defect was created at the mesial root, while a groove was made on the buccal area from the top of the teeth to the bottom of the defect, exposing the dental pulp. The mesial roots of P2, P3 and P4 were extracted 1 month after the induction of the dehiscence defect with chronic pathology. Three teeth were randomly allocated to these experimental groups: (i) spontaneous healing without any bone graft (Control group: C); (ii) ridge augmentation with β-tricalcium phosphate (β-TCP) granules (Test 1 group: T1); and (iii) ridge augmentation with 60% hydroxyapatite (HA) and 40% β-TCP microspheres (Test 2 group: T2). Postmortem histopathologic examination showed significant between-group differences in C and T1 and C and T2 in bone volume/tissue volume in qualitative micro-computed tomography (CT) analysis, as well as significant intergroup differences in the coronal area at 4 and 12 weeks. The composition of connective tissue and mineralized bone in C and T1 were higher than in T2 at 4 weeks of healing, whereas the composition of mineralized bone was higher in T2 than in T1 at 12 weeks of healing. Biphasic calcium phosphate, composed of 60% HA and 40% β-TCP microsphere (i) potentially prevented marked osteoclastic resorption and (ii) promoted ridge preservation in the extraction socket with the dehiscence defect and chronic pathology.

## 1. Introduction

Appropriate treatment for the extraction site before implant placement is dependent upon understanding the biological healing processes involved in the extraction socket. Several studies have reported that alveolar contractions occur horizontally and vertically after tooth extraction [1,2,3].

Most of these studies have investigated dimensional changes after tooth extraction in the intact extraction socket wall and/or with periodontally healthy status. However, tooth extractions in the clinical scenario are undertaken in the setting of periodontitis and/or endodontic–periodontic combined lesions. Recent studies have reported that the extraction socket of infected teeth may exhibit delayed healing, or heal with fibrous scar tissue [4,5].

Ridge preservation can significantly reduce ridge contraction, as compared to sites that have spontaneously healed. Moreover, ridge augmentation to reduce ridge contraction can be carried out when the extraction socket wall is damaged [6,7]. However, the effect of the flapless ridge preservation/augmentation and/or extraction socket healing at the infected site could not be ascertained in the model used in the abovementioned studies.

On the other hand, the infected extraction socket model has been used in various literature [8,9,10,11]. However, those induced infection models have not been standardized with regard to the extent of bone destruction, which poses challenges to comparisons of extraction socket management. A previous study found that sites with induced infection showed irregular bone-destruction patterns, and the extent of destruction could not be standardized [9]. Therefore, we considered that it is necessary to investigate the implementation of an infected tooth and bone defect model, with standardized size and shape, in beagle dogs.

In the alveolar regeneration or reconstruction, autogenous graft, xenograft, allograft and alloplast have been suggested as biomaterials [12]. Among them, synthetic bone substitutes, including beta-tricalcium phosphate (β-TCP) and hydroxyapatite, are known allografts for ridge preservation [13,14]. Both β-TCP and biphasic calcium phosphate (a mixture of β-TCP and hydroxyapatite) have been recommended as bone graft substitutes in various clinical studies [15,16]. However, it is controversial whether resorbable β-TCP could be used in ridge preservation to maintain the alveolar ridge, compared to ridge preservation with biphasic calcium phosphate.

This study was undertaken to comparatively examine spontaneous healing and ridge augmentation in surgically-created dehiscence defects with chronic pathology in dogs. Moreover, we aimed to create an experimental model of a damaged extraction socket wall through surgical resection of the buccal wall of the teeth, and induced chronic pathology to carry out flapless ridge augmentation that was more clinically relevant to the extraction situation.

## 2. Materials and Methods

### 2.1. Experimental Preparation

This animal study was approved by the Institutional Animal Care and Use Committee of Seoul National University (IACUC; approval no. SNU-170123-1-1), and conducted in accordance with the ARRIVE guidelines from 2017 to 2018. Six beagle dogs (male; weight: ~10 kg; age: 8–9 months) procured from OrientBio (Seoul, Korea) were used in this study. The experimental schedule is shown in Figure 1.

### 2.2. Induction of Dehiscence Defect with Chronic Pathology

General anesthesia was induced by intravenous injection of Zoletil (250 mg tiletamine hydrochloride in powder and 250 mg zolazepam hydrochloride in powder is mixed with 5 mL of solvent; 0.1 mg/kg; Virbac, Carros, France), xylazine (2.3 mg/kg; Rompun, Bayer Korea, Ansan, Korea), and atropine sulfate (0.05 mg/kg; Jeil, Daegu, Korea). Thereafter, local anesthesia was induced by injecting 1:100,000 epinephrine-containing lidocaine (Huons, Sungnam, Korea). Intracrevicular incisions were made in the mandibular premolar area, and the flaps were elevated (Figure 2a,d). The mandibular second, third and fourth premolars (P2, P3, and P4) were hemi-sectioned by using a diamond burr (Figure 2c). The pulp of the distal root was extracted with a 25 mm K-file (#15 and #20, MANI, Inc., Utsunomiya, Tochigi, Japan), and obturated with a gutta-percha master cone using cold condensation with accessory cones of AH Plus (Dentsply, DeTrey GmbH, Konstanz, Germany). After the root canal treatment, the distal root was sealed with intermediate restorative material (IRM; Dentsply Sirona, Milford, DE, USA; Figure 2c). A dehiscence defect, measuring 3 mm in width and 4 mm in height, was created at the mesial root, and a groove on the buccal area from the top of the teeth to the bottom of the defect was also made, thereby exposing the dental pulp (Figure 2d,e). The surgical site was closed with 4/0 Vicryl (Figure 2f). Suture removal was done 1 week after the surgery.

### 2.3. Tooth Extraction and Ridge Augmentation

The mesial roots of P2, P3 and P4 were extracted 1 month after the induction of the dehiscence defect with chronic pathology. Three teeth were randomly allocated to the following experimental groups: (i) spontaneous healing without any bone graft (Control group: C); (ii) ridge augmentation with β-TCP granules (Cerasorb, Curasan AG, Kleinostheim, Germany; Test 1 group: T1); and (iii) ridge augmentation with 60% hydroxyapatite (HA) and 40% β-TCP microspheres (HansBiomed, Seoul, Korea; Test 2 group: T2). The granulation tissue was removed thoroughly with a surgical curette after tooth extraction. We applied a double layer of resorbable, non-crosslinked collagen membrane (BioGide^®^, Geistlich Biomaterials, Wolhusen, Switzerland) at the ridge augmentation site, as previously described [13]. The surgical site was closed with 4/0 Vicryl, and the flap was not mobilized to achieve a primary closure (Figure 3).

### 2.4. Postoperative Protocol

To alleviate any postoperative trauma, the subjects were provided soft food, and analgesics were administered to reduce postoperative pain. A plaque control protocol was carried out every fortnight after suture removal.

### 2.5. Micro-CT Analysis

At 4 and 12 weeks after the ridge augmentation, the animals were sacrificed, and the experimental sites were resected out as specimens. After fixation in a tissue block, we obtained micro- computed tomography (CT) images of the surgical sites with the dehiscence defect (SkyScan-1173, Kontich, Belgium). The radiation-exposure conditions for micro CT imaging were as follows: 130 kV, 60 μA, 500 ms, 1.0 mm aluminum filter, image pixel size 19.18, rotation step 0.300 degree. The scanned data were reconstructed into three-dimensional images using a CT analysis software program (CT Analyzer version 1.17, Skyscan, Kontich, Belgium).

### 2.6. Qualitative Micro-CT Analysis

Image analysis was conducted to measure the bone morphometric parameters for each volume of interest (VOI), and the VOI of the present study was determined to correspond to the extraction socket of the mesial root of P2, P3 and P4. All units of dimensions are specified as parameters according to the specifications recommended by the American Society for Bone and Mineral Research Histomorphometry Nomenclature Committee [17]: bone volume (BV), tissue volume (TV), volume density (BV/TV), bone surface (BS) and surface to volume ratio (BS/BV). The qualitative micro-CT analysis was conducted as BV/TV and BS/BV.

### 2.7. Quantitative Analysis of the Superimposed Micro-CT Images

The residual root of each distal site was used as a reference alveolar ridge when calculating the ridge contraction. To investigate the change after tooth extraction and/or ridge augmentation, each middle section of the mesial root was superimposed with reference sections (distal root) as previously described [6,11]. The area was demarcated into three regions at each superimposed image: coronal 1/3, middle 1/3 and apical 1/3, and then the dimensional changes of the experimental area compared to the reference area were calculated and expressed as percentages in these three areas (Figure 4).

### 2.8. Histologic Processing

The biopsy block was fixed in formalin for 2 weeks, and then demineralized in 5% formic acid for 10 days before being embedded in paraffin. The blocks were sectioned with 3-µm slice thickness bucco-lingually, paralleling the long axis of the extraction site and stained with hematoxylin and eosin. The middle sections of the tissue specimens of each extraction site were used for histologic analysis.

We conducted histologic analysis by using an incandescent light microscope (DP72; Olympus, Tokyo, Japan) and an imaging system (DP Controller; Olympus). A periodontist (J.L.) carried out histologic and histomorphometric analysis. The ratio of connective tissue area, graft material area, and mineralized bone area was determined in an image measurement program (Tomoro Scope Eye version 3.6.6, Saramsoft Co., Ltd., Anyang-si, Gyeonggi-do, Korea), as previously reported [13].

### 2.9. Statistical Analysis

Statistical analyses were conducted with the SPSS version 17 software (IBM Software, Armonk, NY, USA). To verify the intergroup differences among the different parameters, the Mann–Whitney and Kruskal–Wallis tests were used for two and three group means, respectively. When the outcomes of the Kruskal–Wallis test were significant, Bonferroni multiple comparison was performed using the Mann–Whitney test (0.05/3 ≒ 0.017).

## 3. Results

### 3.1. Clinical Findings

Soft-tissue healing was complete at 1-month post-induction of the dehiscence defect with chronic pathology (Figure 3a). Dental plaque accumulated around the surgically created notch area of the mesial root, with resultant inflammation characterized by erythema and edema. When the tooth with chronic pathology was extracted, a marked buccal bone defect was detected by using a probe compared to the lingual bone plate. All extraction sockets in this study showed uneventful healing without any inflammation after the healing period of spontaneous healing/ridge augmentation.

### 3.2. Qualitative Micro-CT Analysis

The values recorded by using micro-CT are presented in Table 1. The BV/TV values were 27.78% ± 1.28%, 33.24% ± 6.07% and 34.38% ± 6.72%, as well as 33.75% ± 4.73%, 46.40% ± 5.17% and 53.12% ± 10.73% in the C, T1, and T2 groups at the 4- and 12-week observation timepoints, respectively; there were no statistically significant intergroup differences at the 4-week evaluation, but we detected significant differences between the C and T1 groups and C and T2 groups at the 12-week evaluation. Furthermore, BV/TV values tended to gradually increase from the 4- to 12-week observation in intragroup comparisons (Table 1).

The BS/BV values were 14.23 ± 0.62, 12.42 ± 2.68 and 13.38 ± 2.84 mm^−1^, along with 8.71 ± 0.36, 6.69 ± 1.58 and 8.67 ± 1.92 mm^−1^ in the C, T1 and T2 groups at the 4- and 12-week observations, respectively. There were no significant differences among the three groups at the 4- and 12-week assessments (Table 1); however, BS/BV values tended to gradually decrease from the 4- to 12-week observation on intragroup comparisons (Table 1).

### 3.3. Quantitative Analysis of the Superimposed Micro-CT Images

The results of quantitative micro-CT analyses at coronal, middle and apical areas is presented in Table 2. In the 4-week observation, the dimensional proportions at the coronal area were 62.72% ± 6.11%, 77.40% ± 9.96% and 103.07% ± 14.85% in the C, T1 and T2 groups, respectively, with statistically significant differences on intergroup comparisons.

The dimensional proportions in the middle and apical areas were 90.72% ± 11.16%, 93.56% ± 3.70% and 96.58% ± 2.38%, along with 98.62% ± 2.68%, 98.37% ± 6.52% and 97.67% ± 2.83% in the C, T1 and T2 groups, respectively, and there were no significant intergroup differences at the middle and apical areas.

At the 12-week observation, the dimensional proportions at coronal area were 54.11% ± 2.10%, 57.65% ± 7.25% and 96.75% ± 4.61% in C, T1 and T2 groups, respectively. The value in group T2 was greater than the value in groups C and T1. The dimensional proportions at the middle area were 82.96% ± 11.68%, 89.89% ± 5.85% and 91.23% ± 6.22% in C, T1 and T2 group, respectively. The dimensional proportion at the apical area were 96.78% ± 4.14%, 94.81% ± 6.72% and 96.72% ± 8.73% in the C, T1 and T2 groups, respectively. There were no statistically significant differences among these three groups at the middle and apical areas.

### 3.4. Histologic Analyses

Uneventful extraction socket healing was observed, and alveolar bone ridge resorption was remarkable in the coronal area in the control group. The formation of woven bone was observed at the 4-week assessment, and the bone-maturation process, including cortical bone and bone marrow formation, was noted at the 12-week observation (Figure 5a and Figure 6a).

No specific inflammatory findings were found in the T1 group. Bone graft materials persisted at the 4-week observation, but had been absorbed and replaced with bone at the 12-week observation (Figure 5b and Figure 6b), and there was woven bone formation in the extraction socket similar to the bone picture in the control group at the 4-week observation. Overall, bone healing was considered to have progressed well without any inflammatory event.

No specific inflammatory findings were seen in the T2 group. Most bone grafts persisted up to 12 weeks, and mineralized bone tissue was formed within the bone graft material (Figure 5c and Figure 6c). Bone neoformation appears to be expanded from the inner area of the graft at the 4- and 12-week observations. No cortical bone formation appeared at either the 4- or 12-week observation in the buccal and coronal areas, which was the bone defect area of chronic pathology.

### 3.5. Histomorphometric Analysis

The outcome of histomorphometric analysis is presented in Table 3. At the 4-week observation, we noted 46.86% ± 6.26% of connective tissue and 53.14% ± 6.26% of mineralized tissue in the C group. In T1, we observed 40.10% ± 4.90% of connective tissue, 8.28% ± 2.07% of bone particles and 51.62% ± 4.94% of mineralized bone. In T2, we observed 34.65% ± 2.74% of connective tissue, 34.11% ± 3.17% of bone particles and 31.24% ± 2.70% of mineralized bone. The composition of connective tissue and mineralized bone in the C and T1 groups showed higher values compared to that in T2. However, the composition of bone particles in T2 showed higher values compared to that in T1.

At the 12-week observation, in the control group, we observed 48.66% ± 9.02% of connective tissue and 51.34% ± 9.02% of mineralized tissue. In T1, we measured 40.86% ± 4.81% of connective tissue, 0.00% ± 0.00% of bone particles and 59.14% ± 4.81% of mineralized bone. In T2, we quantified 21.82% ± 5.54% of connective tissue, 34.99% ± 1.71% of bone particles and 43.19% ± 5.03% of mineralized bone. Interestingly, bone particles were resorbed in T1 at the 12-week observation. The composition of connective tissue in C and T1 showed higher values compared to that in T2. There was a higher composition of bone particles and mineralized bone in T2 compared to that in T1.

## 4. Discussion

The present study investigated the effect of ridge augmentation using the β-TCP and biphasic calcium phosphate sphere in the extraction the socket with dehiscence defect and chronic pathology. In contrast to the marked resorption during tissue modeling and remodeling in ridge augmentation with β-TCP and the control group, we found no significant changes in ridge augmentation in the biphasic calcium phosphate sphere group. Thus, ridge augmentation with the biphasic calcium phosphate sphere seems to counteract ridge resorption following the extraction of compromised teeth.

The qualitative micro-CT analysis of images in this study showed no significant intergroup differences for BV/TV and BS/BV at 4 weeks of healing, although BV/TV showed significant between-group differences between the C and T1 or C and T2 groups at 12 weeks of healing. BV/TV is an indicator of the ratio of the trabecular bone volume to the total VOI [18]. However, this BV/TV ratio in the ridge augmentation group did not differ significantly compared with the control group at 4 weeks, but differences were significant at 12 weeks. This result is due to immature trabecular bone formation at 4 weeks, whereas mature trabecular bone formation occurs by 3 months. Moreover, it takes approximately 3 months for bone maturation in the tooth-extraction or ridge-preservation sites [6,19]. On the other hand, BS/BV, an indicator of bone complexity [20], did not show significant intergroup differences in contrast to a previous study that undertook ridge preservation using a non-resorbable xenograft [10]. A possible explanation for this observation is the inherent differences of the materials used in ridge preservation/augmentation. In the present study, we used resorbable biomaterials containing 100% β-TCP (T1), or biphasic phosphate containing 40% β-TCP. The use of these fully- or partially-resorbable bone grafts in the present study ensured that the proportion of encapsulated grafts was relatively lesser than in previous studies, which eventually resulted in the absence of significant intergroup differences in bone complexity.

The quantitative micro-CT of this study showed intergroup differences in the coronal area at 4 and 12 weeks of healing, which was significant for C and T1 at 4 weeks, but not at 12 weeks of healing. This finding might have resulted from the rapid resorption of the β-TCP. The result of the present study is in agreement with findings from a previous study that evaluated the influence of bone graft materials on ridge preservation [21]. In contrast, the coronal area in T2 showed lesser contraction compared with the C and T1 groups. Using non-resorbable hydroxyapatite materials, the ridge volume seems to be maintained for up to 3 months of healing as previously reported [13,22].

The study tissue specimens showed that the connective tissue and mineralized bone ratio was lower in T2 compared with the C or T1 groups at 4 and 12 weeks of healing. In addition, a newly formed bone bridge from the buccal and lingual bone crests was observed at 4 weeks of healing in the control group. These findings are similar to results from a previous study, which demonstrated that non-grafted extraction sockets after 4 weeks of healing had become completely filled with woven bone [1,19]. In contrast to the control group, we observed less woven bone formation adjacent to the socket wall, including adjoining graft materials in the T2 group within the extraction socket at 4 weeks of healing. These results indicate that wound healing and bone remodeling could be influenced or retarded by alloplastic bone grafts, which is similar to results from previous studies with various materials [23,24,25]. The ratio of mineralized bone occupied in the extraction sockets increased from 31% to 43% in the T2 group, whereas the ratio of bone particle remained more or less stable at approximately 34%. This trend is in agreement with findings from animal studies which undertook ridge preservation with other osteoconductive bone grafting materials [13,23,25].

Recently, various studies have been published on extraction sockets with periodontal defects. The initial models were presented in the literature with surgically-created defects, which were far from clinical conditions, in that these models had no chronic pathology due to the absence of bacterial infection [6,7]. The next model suggested in the literature was the endodontic–periodontic combined lesion model, which was more advanced, in that it reflected an inflammatory condition [9,10]. However, there was a limitation because the bone destruction pattern did not show a standardized defect size for each site and animal.

The extraction sockets with dehiscence defect and chronic pathology model in this study seem appropriate to study flapless ridge augmentation in compromised teeth, because it has a standardized defect and inflammatory state that reflects the clinical situation associated with tooth extraction.

In this study, according to previous studies [10,11], the effect of ridge preservation/augmentation on the mesial root was compared with the distal root. However, since it is better to superimpose the identical location anatomically than to superimpose the teeth at a different location, future studies should be conducted in consideration of this point of view.

The reduction at the coronal area in the control group was approximately 50% at 12 weeks of healing, which is in agreement with the result of extraction socket healing without a dehiscence defect in previous studies [1,13]. This result suggests that the dehiscence defect might not be a critical defect that reduces hard tissue filling in the extraction socket. Buccal bone is naturally resorbed following the creation of a fresh extraction socket [19], because the buccal wall is thin. This biological characteristic might counteract the effect of a dehiscence defect on hard tissue healing. Future studies in a more severe bone defect model, including lingual bone defects, should be considered to verify the impact of bone defects on hard tissue healing. In addition, clinical studies with the systematic classification of bone defects with chronic pathology around teeth should be investigated to identify the effect of flapless ridge preservation/augmentation.

## 5. Conclusions

Biphasic calcium phosphate, composed of 60% HA and 40% β-TCP microsphere (i) possibly prevented marked osteoclastic resorption and (ii) promoted ridge preservation in the extraction socket with a dehiscence defect and chronic pathology. Flapless ridge augmentation with biphasic calcium phosphate with a non-crosslinked collagen membrane may be suggested when the extraction socket exhibits buccal dehiscence with chronic pathology.

## Figures and Tables

**Figure 1 materials-13-01452-f001:**
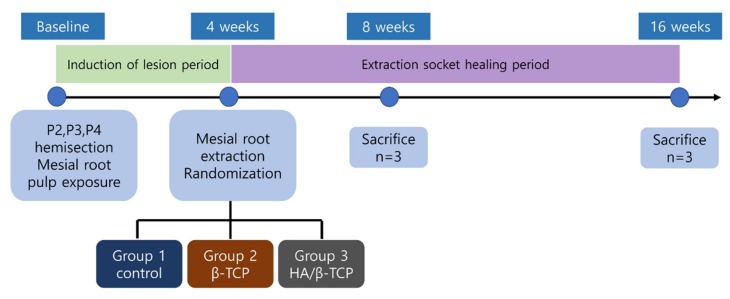
Study flowchart. β-TCP: β-tricalcium phosphate, HA: hydroxyapatite.

**Figure 2 materials-13-01452-f002:**
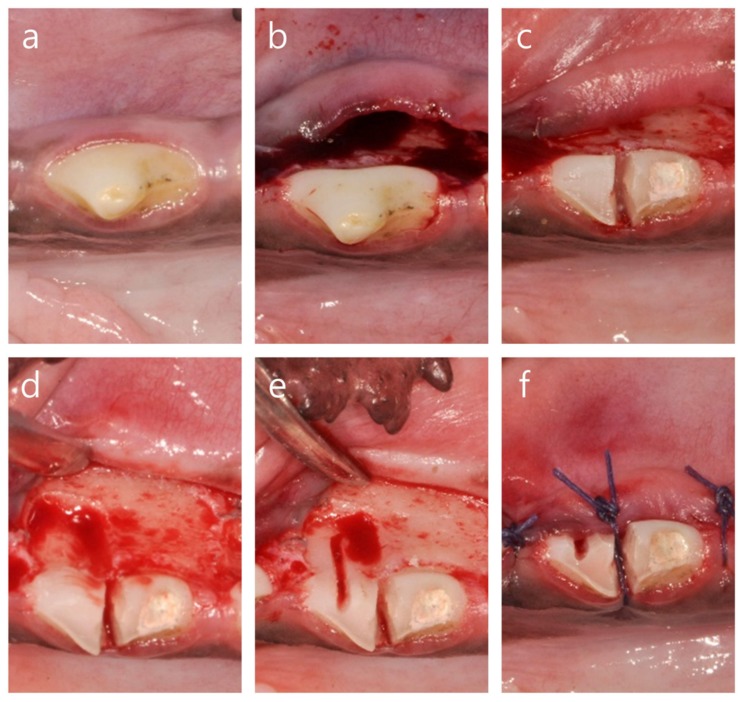
Clinical photographs of the induction of the dehiscence defect with chronic pathology. (**a**) Preoperatively, (**b**) flap elevation, (**c**) hemi-section and distal root canal treatment, (**d**) creation of a dehiscence defect on the buccal side of the mesial root, (**e**) a vertical notch on the buccal side with exposure of the pulp of the mesial root, and (**f**) flap suture.

**Figure 3 materials-13-01452-f003:**
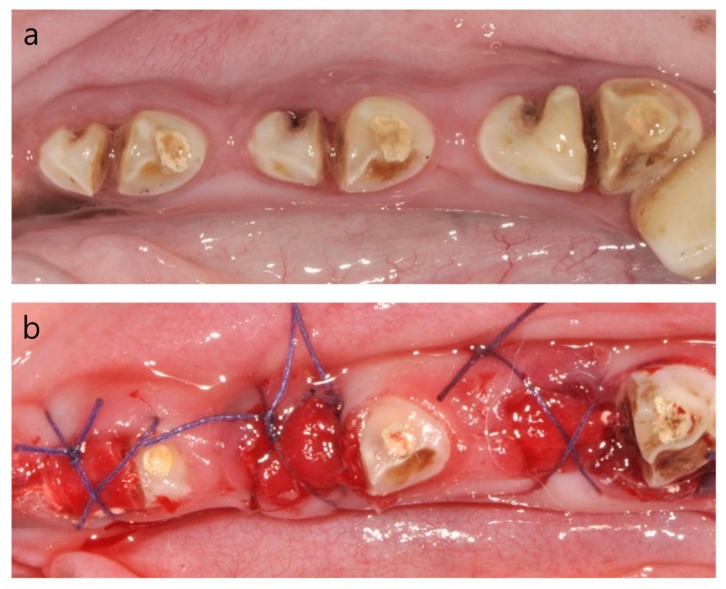
Clinical photographs (**a**) before extraction and (**b**) after ridge preservation.

**Figure 4 materials-13-01452-f004:**
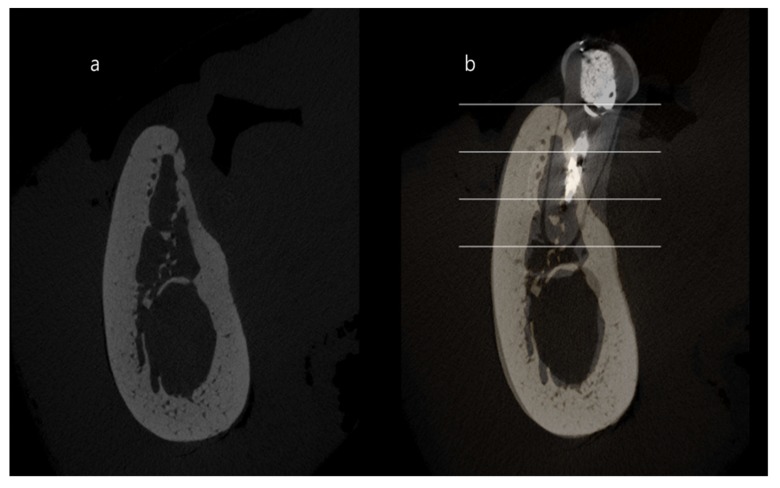
Buccolingual micro-computed tomography images. (**a**) extraction socket of mesial root area and (**b**) superimposition of extraction socket (mesial) area and distal root. The area was divided into three areas, with coronal, middle and apical regions.

**Figure 5 materials-13-01452-f005:**
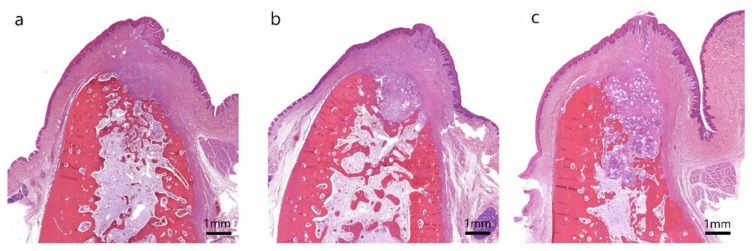
Buccolingual section on hematoxylin and eosin staining at 4 weeks after extraction in the (**a**) control group, (**b**) T1 group and (**c**) T2 group.

**Figure 6 materials-13-01452-f006:**
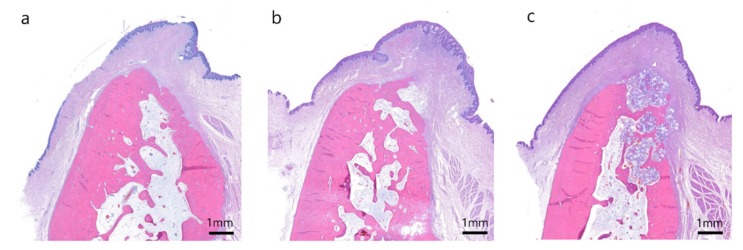
Buccolingual section on hematoxylin and eosin staining at 12 weeks after extraction in the (**a**) control group, (**b**) T1 group and (**c**) T2 group.

**Table 1 materials-13-01452-t001:** Results of qualitative micro- computed tomography (CT) analyses at 4 and 12 weeks of healing.

Healing time	Parameters	Control	T1	T2	*p*-Value
4 weeks	BV/TV (%)	27.78 ± 1.28	33.24 ± 6.07	34.38 ±6.72	0.051
-	BS/BV (1/mm)	14.23 ± 0.62	12.42 ± 2.68	13.38 ± 2.84	0.551
12weeks	BV/TV (%)	33.75 ± 4.73 ^a^	46.40 ± 5.17^b^	53.12 ±10.73 ^b^	<0.001
-	BS/BV (1/mm)	8.71 ± 0.36	6.69 ± 1.58	8.67 ± 1.92	0.142
-	-	-	-	-	-

* The Kruskal–Wallis test was used to determine differences among the groups at a significant level of α < 0.05. Different letters, ^a^, and ^b^ indicate statistical differences under Bonferroni correction (overall *p*-value 0.05). BV, Bone volume; TV, Total volume; BS, Bone surface.

**Table 2 materials-13-01452-t002:** Results of the quantitative micro-CT analyses of coronal, middle and apical area at 4 and 12 weeks of healing.

Healing Time	Dimensional Proportion	Control	T1	T2	*p*-Value
4 weeks	Coronal area	62.72 ± 6.11 ^a^	77.40 ± 9.96 ^b^	103.07 ± 14.85 ^c^	<0.001 ^*^
-	Middle area	90.72 ± 11.16	93.56 ± 3.70	96.58 ± 2.38	0.505 ^*^
-	Apical area	98.62 ± 2.68	98.37 ± 6.52	97.67 ± 2.83	0.608 ^*^
12 weeks	Coronal area	54.11 ± 2.10 ^a^	57.65 ± 7.25 ^a^	96.75 ± 4.61 ^b^	<0.001 ^*^
-	Middle area	82.96 ± 11.68	89.89 ± 5.85	91.23 ± 6.22	0.418 ^*^
-	Apical area	96.78 ± 4.14	94.81 ± 6.72	96.72 ± 8.73	0.949 ^*^

* The Kruskal–Wallis test was used to determine differences among the groups at a significant level of α < 0.05. Different letters, ^a^, ^b^ and ^c^ indicate statistical differences under the Bonferroni correction (overall *p* -value = 0.05).

**Table 3 materials-13-01452-t003:** Results of the histomorphometric analyses at 4 and 12 weeks of healing.

Healing time	Composition	Control	T1	T2	*p*-Value
4 weeks	Connective tissue	46.86 ± 6.26 ^a^	40.10 ± 4.90 ^a^	34.65 ± 2.74 ^b^	0.001 *
-	Bone particle	-	8.28 ± 2.07	34.11 ± 3.17	0.002 ^#^
-	Mineralized bone	53.14 ± 6.26 ^a^	51.62 ± 4.94 ^a^	31.24 ± 2.70 ^b^	<0.001 *
-	-	-	-	-	-
12 weeks	Connective tissue	48.66 ± 9.02 ^a^	40.86 ± 4.81 ^a^	21.82 ± 5.54 ^b^	<0.001 *
-	Bone particle	-	0.00 ± 0.00	34.99 ± 1.71	0.002 ^#^
-	Mineralized bone	51.34 ± 9.02 ^a,b^	59.14 ± 4.81 ^a^	43.19 ± 5.03 ^b^	0.002 *

* The Kruskal–Wallis test was used to determine differences among three groups at a significant level of α < 0.05. ^#^ The Mann–Whitney test was used to determine difference between two groups at a significant level of α < 0.05. Different letters, ^a^ and ^b^, indicate statistical differences under the Bonferroni correction (overall *p*-value = 0.05).

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
