# Peer review of "Ridge Augmentation Using β-Tricalcium Phosphate and Biphasic Calcium Phosphate Sphere with Collagen Membrane in Chronic Pathologic Extraction Sockets with Dehiscence Defect: A Pilot Study in Beagle Dogs"

_materials, 2020, doi:10.3390/ma13061452_

Round 1

Reviewer 1 Report

The manuscript topic is actual and the paper has merit. It could be attractive, adequate and interesting for the journal readers. However there are some point that authors should address in order to have a final more structured paper.

Firstly Authors should underline the limitation of the value of the study, and the clinical and surgical implication of the presented study should be added. At this stage the paper seems to be directed to not clinical or surgeons readers. Please emphasize the clinical application of the study.The limitation of an "animal study" should be underlined and need to be synthesized in a paragraph as follows "....animal studies will only become more valid predictors of human reactions to exposures and treatments if there is substantial improvement in both their scientific methods as well as in more systematic review of the animal literature as it evolves. Systematic reviews of animal research, if they are used to inform the design of clinical trials, particularly with respect to appropriate drug dose, timing and other crucial aspects of the drug regimen, will further improve the predictability of animal research in human clinical trials...."

Introduction section should highlights the clinical rationale of this paper. Otherwise the study seems to be directed to just scientist or researcher and not to surgeons. References are inadequate. Introduction section is poor. Some more references about the recent (2015-2020) CLINICAL reconstructive option or about biomaterials just published should be added. as sample;

Cicciù, Marco et al. “Facial Bone Reconstruction Using both Marine or Non-Marine Bone Substitutes: Evaluation of Current Outcomes in a Systematic Literature Review.” Marine drugs vol. 16,1 27. 13 Jan. 2018, doi:10.3390/md16010027

Author Response

Dear Reviewer

Please confirm the attached file.

Sincerely yours,

Reviewer 2 Report

The study present a interesting results, especially when the uCT was performed after 4 and 12 weeks material implantation. Usually, this investigation is carry out after 13 weeks.So, the results after 4 weeks are also valuable and informative. In my opinion, the manuscript should be enrich with the data about the materials - more information about the composition of the materials and their properties. However, the in vivo part is readable and the methodology was appropriate presented and applied. 

Reviewer 3 Report

I would like to congratulate the authors on their work. The buccal dehiscence defect with chronic inflammation used in this study makes the conclusions drawn even more applicable to daily clinical practice.

According to the results of the present study the use of β-TCP results in insufficient volumetric maintenance of the original alveolar contours. However, due to its osteoconductive properties bone formation is increased after 12 weeks until which time bone graft particles undergo total resorption. The β-TCP content of BCP is resorbed, the HA content on the other hand integrates in the grafted socket. This results in volumetric stability making BCP a suitable material for alveolar ridge preservation (ARP) from space maintenance point of view. However, non-resorbable HA content of the BCP inhibits natural bone healing. Bone bridging between the buccal and lingual cortical which is present in the case of the ungrafted socket and delayed using β-TCP is missing when the socket is grafted by BCP. For the clinician it is a concern that the marginal bone adjacent to the neck of the implant which is most vulnerable to periimplantitis consists of bone graft material encapsulated by connective tissue. However, literature reviews found no evidence that on the long term implants placed in grafted sockets show increased marginal bone loss compared to implants placed in non-grafted sockets.

A slight weakness of the study that volumetric measurements following ARP are compared to the anatomic location close to the test or control socket but not identical. Another weakness of the study might be that the histomorphometric results reported are of the middle 1/3rd of the sockets. The literature suggests that the percentage of tissues of the sockets are different in the coronal, middle and apical 1/3rd of the sockets.

I find the surgical methods very well illustrated and easy to understand. However, at the “Tooth extraction and ridge augmentation” section I suggest including in the text, that the flap was not mobilized to achieve a primary closure, because there are several surgical procedures described in the literature for a successful alveolar ridge preservation.

In row 221 there is a reference to Table 3. However, it is missing. Please insert.

In row 250 Citation 10 should be in brackets.

Author Response

(The authors gave the same response as above.)

Round 2

Reviewer 1 Report

Authors made excellent job addressing all the reviewers request and note